# Evaluation of the Risk of Anxiety and/or Depression during Confinement Due to COVID-19 in Central Spain

**DOI:** 10.3390/ijerph18115732

**Published:** 2021-05-27

**Authors:** Rosa M Cárdaba-García, Lucia Pérez Pérez, Virtudes Niño Martín, Inés Cárdaba-García, Carlos Durantez-Fernández, Elena Olea

**Affiliations:** 1Nursing Department, Faculty of Nursing, University of Valladolid, 47005 Valladolid, Spain; rosacardaba@yahoo.es (R.M.C.-G.); lperezp@saludcastillayleon.es (L.P.P.); vninoger@gmail.com (V.N.M.); olea@ibgm.uva.es (E.O.); 2Nursing Care Research (GICE), Faculty of Nursing, University of Valladolid, 47005 Valladolid, Spain; 3Emergencies Management (SACYL), 40002 Segovia, Spain; 4Primary Care Management Valladolid West (SACYL), 47012 Valladolid, Spain; 5Primary Care Management Valladolid East (SACYL), 47010 Valladolid, Spain; 6General Hospital of Segovia (SACYL), 40002 Segovia, Spain; inescardaba@gmail.com; 7Faculty of Health Sciences, University of Castilla-La Mancha, 45600 Talavera de la Reina, Spain; 8Department of Biochemistry and Molecular Biology and Physiology, University of Valladolid, 47005 Valladolid, Spain

**Keywords:** SARS-CoV-2, pandemic, confinement, Spain, depression, anxiety

## Abstract

(1) Background: The confinement of the population in response to the COVID-19 pandemic was related to an increased risk of suffering from anxiety and/or depression in previous studies with other populations. (2) Methods: descriptive study using surveys (Goldberg Anxiety and Depression Scale) with 808 participants over 18 years of age between 14 and 20 of May 2020 during the confinement due to the SARS-CoV-2 virus in Spain. (3) Results: 63% of the participants were at risk of suffering from anxiety and 64.9% were at risk of depression. Variables reaching statistical significance were: age (t anxiety = −0.139 and t depression = −0.153), gender (t anxiety = −4.152 and t depression = −4.178), marital status (anxiety F = 2.893 and depression F = 3.011), symptoms compatible with COVID-19 (t anxiety = −4.177 and t depression = −3.791), previous need for psychological help (t anxiety = −5.385 and t depression = −7.136) and need for such help at the time of the study (t anxiety = −9.144 and depression = −10.995). In addition, we generated two regression models that estimate the risk of anxiety and depression. (4) Conclusions: more than half of the participants were at risk of suffering from anxiety and/or depression, confirming the negative effect of confinement on the population.

## 1. Introduction

The pandemic produced by the SARS-CoV-2 virus has generated a global emergency, which has led to a series of measures that have included the confinement of the population in Spain and other countries [1,2].

On 31 December 2019, the WHO received a statement from the Chinese authorities of several cases of atypical pneumonia in the city of Wuhan [2,3]. A week later, this disease was termed COVID-19 (coronavirus disease 2019) after it was discovered that it was caused by a new coronavirus [4]. Months later, the virus spread rapidly throughout the world, with Spain being one of the most affected countries [5]. For this reason, the Spanish Government published a Royal Decree 463/2020 [6] on March 14 in the Official State Gazette, declaring a state of alarm for the management of the health crisis caused by COVID-19. Among other measures, the decree obliged the population to confine themselves to their homes from 15 March to 3 May 2020 [7]. This restriction of freedom of movement with the aim of controlling viral transmission was maintained at the time of the study, despite the relaxation of some of these restrictions. It was not until the publication of Order SND/380/2020, of 30 April, that the confinement measures were somewhat relaxed for the population over 14 years of age to allow outdoor sports [8]. Therefore, the Spanish population had remained strictly isolated in their homes for a little less than two consecutive months.

Although containment measures were necessary to control the outbreak, the possible consequences that confinement may have on the mental health of the population are still worrying [9,10,11]. Even the WHO recommended that the impact of confinement on people be quickly assessed to implement timely health measures that mitigate its harmful effects [12]. No similar situation has existed in Europe in recent years, but preliminary studies provide evidence of the deterioration of mental health in the population due to confinement. In China, two studies show higher rates of anxiety, depression, alcohol consumption and a lower proportion of mental well-being than usual in the population during the pandemic [13,14]. A study in India, with 1000 respondents with the Depression Anxiety Stress Scales (DASS21) questionnaire, presents significant differences between depression, anxiety and stress according to age, gender and employment [15]. In addition, in Italy, the mental health of the population has been affected and yet there have been fewer hospital admissions for psychological causes [16,17]. In the United Kingdom, research has been conducted that predicts anxiety and depression as a function of the presence of low income or loss of income and pre-existing health conditions in self and others [18]. Similarly in Denmark, higher rates of anxiety and depression due to confinement have also been found [19]. In Spain, the geographical area most affected by COVID-19 disease has been the center of the country [20], however, there is not previous research specifically investigating the mental health of the confined population in this region.

According to the prevalence of anxiety and depression in the aforementioned studies, we propose as working hypothesis that the confinement of the population of Spain was related to a greater risk of anxiety and/or depression. Hence, the objective of this study was to determine the risk of suffering from anxiety and/or depression in the Spanish population confined to regions of central Spain due to COVID-19.

## 2. Materials and Methods

### 2.1. Study Design

This is a descriptive study with a survey methodology carried out on a confined population. The data were collected from 14 to 20 May 2020. At that moment, the population had been confined to their homes for two months and de-escalation in Phase 0 had begun throughout Spain.

The study included persons residing in the central region of Spain during confinement, over 18 years of age, Spanish-speaking, with sufficient technical skills to answer an online survey and who voluntarily consented to participate in the study. We excluded Spaniards residing in other geographic areas than the central part of the country, those under 18 years of age, individuals who do not speak Spanish, those who don´t have the ability to handle online questionnaires and those who checked the refusal box in the study participation, even though they had completed the questionnaire.

### 2.2. Population and Ethical Aspects

We included the entire population over 18 years of age residing in central regions of Spain during the state of alarm due to SARS-CoV-2, who voluntarily accepted participation in the study on the indicated dates. A sample of 808 people who met the inclusion criteria was obtained.

The study was approved by the Ethics Committee for Drug Research of the Valladolid East health area, with registration code PI 20-1803 NO HCUV on 14 May 2020. This study conforms to the STROBE Initiative (Strengthening the Reporting of Observational Studies in Epidemiology) for observational studies of the EQUATOR Initiative [21].

### 2.3. Outcome Measures

The study variables we aimed to describe and correlate were: age, gender, marital status, being a health worker, place of confinement, number of confined people in the home, having been dismissed from work, suffering from COVID-19 symptoms (fever, headache, dry cough, sore throat, dyspnea, nausea and vomiting, diarrhea, fatigue, ageusia, and anosmia) [16], undergoing COVID-19 diagnostic tests (rapid antigen detection test or viral RNA detection by RT-PCR), a family member or friend who suffered from COVID-19, previous need for mental healthcare, current need for mental healthcare, being at risk of suffering from anxiety and/or depression.

### 2.4. Data Collection

Due to the conditions at the time of study, sampling was non-probabilistic through volunteering and secondarily by chain referral to obtain the largest possible sample. Recruitment was carried out through social networks (WhatsApp^®^, Facebook^®^ and Twitter^®^), in which the link to the questionnaire was disseminated through Google Forms^®^ on Google Drive^®^.

For data collection, we used a self-administered and anonymous questionnaire in which all the aforementioned variables were collected, using the Goldberg Anxiety and Depression Scale (GADS) [22,23] (Appendix A; Table A1). This screening tool measures the possibility of suffering from anxiety and/or depression. It consists of two subscales with nine items each, whose answers are dichotomous in a Yes/No format, and in which one point is assigned to an affirmative answer and 0 to a negative answer. The first four items of each of the subscales determine the risk of suffering from anxiety (in 4 points) and/or depression (in ≥2 points). Higher scores in each subscale are related to a higher risk, but the total score of the instrument (anxiety score and depression score) is not meaningful and generally not used, unless the first items show alteration. The anxiety scale has a sensitivity of 82% and the depression scale with 85%. The positive predictive value for anxiety is 0.56 and for depression 0.85 [22].

### 2.5. Data Analysis

We used descriptive analysis with measures of sample distribution (frequencies and percentages), centrality (mean) and dispersion (standard deviation [SD]) of the variables. Quantitative variables were analyzed using a normality test (Kolmogorov–Smirnov), using parametric tests (ANOVA, Student’s t-test and Pearson’s correlation coefficient) for the inferential analysis of the results and the total scores of the anxiety and depression subscales.

In an attempt to search for predictive models of factors related to anxiety and depression, we decided to use a Bayesian probability model. A multiple regression analysis was done in successive steps, from which predictive models with significant variables (*p* ≤ 0.01) and model equations were obtained, both for anxiety and depression. For data analysis, we used IBM SPSS Statistics, version 24.0 (SPSS Inc., Chicago, IL, USA). In all tests, a confidence level of 95% and a *p*-value below 0.05 were considered significant.

## 3. Results

### 3.1. Descriptive Data

In total, 808 participants were included, who were between 18 and 80 years old, with a mean of 43.4 years (95% CI: 42.7–44.0, SD 19.0). The vast majority resided in the Autonomous Community of Castilla y León (76.6%); only 85 (10.5%) participants lived alone. The average number of cohabitants in a household was 2.35 (95% CI: 1.9–2.8, SD 1.3). Most participants had kept their jobs, had not been infected with COVID-19, had not presented compatible symptoms or required diagnostic tests. Regarding the need for psychological and psychiatric help, 244 people (30.2%) had needed it at some point in their life, while at the time of study, 10.8% were in need. 31.4% believed that confinement was negatively affecting cohabitation in their homes (Table 1).

Table 2 shows the results of the anxiety and depression subscales in the GADS. To verify the reliability of the results, Crombach’s α was calculated at 0.859 for the total scale (18 items), and at 0.793 and 0.776 for the anxiety (nine items) and depression (nine items) subscales, respectively, which confers validity to the results.

Considering the cutoff number of the first four items of each subscale, 63% (509) of the sample was at risk of suffering from anxiety (4 points in the first four items of the anxiety subscale) and 64.9% (524) were at risk of depression (≥2 in the first four items of the depression subscale) (Table 3). The mean anxiety subscale score was 4.72, (95% CI 3.78–5.66, SD 1.43); while in the depression subscale, the mean of responses was 3.34 (95% CI 2.46–4.22, SD 1.36).

### 3.2. Inferential Analysis

We performed parametric tests after verifying the normal distribution of the total scores on the anxiety and depression subscales to determine the relationship between sociodemographic variables, COVID-19-related variables, and the results of the GADS questionnaire subscales (Table 4).

In relation to age, the Pearson correlation coefficient was −0.139 for anxiety and −0.153 for depression, both weakly statistically significant (*p* = 0.01 bilateral). It is shown that at an older age the risk of suffering from anxiety and depression decreases. Concerning gender differences, we compared means with the Student’s t test, obtaining the value t = −4.152 (*p* = 0.000) in anxiety, and a value t = −4.178 (*p* = 0.000) in depression. Both anxiety and depression symptoms were much more frequent in women.

Among the variables that showed significant results in both anxiety and depression scores, we identified the marital status of the participants (ANOVA anxiety F = 2.893; *p* = 0.002 and ANOVA depression F = 3.011; *p* = 0.002), having had symptoms compatible with COVID-19 (t anxiety = −4.177; *p* = 0.000 and t depression = −3.791; *p* = 0.000) and cohabitation (ANOVA anxiety F = 13.636; *p* = 0.000 and ANOVA depression F = 10.007; *p* = 0.000). Nonetheless, the number of cohabitants did not reach statistical significance according to Pearson’s correlation coefficient, 0.044 for anxiety and 0.027 for depression (*p* = 0.01 bilateral).

In contrast, the variables that did not show a statistically significant relationship were: the geographical place of residence (anxiety ANOVA: F = 0.934; *p* = 0.495 and depression ANOVA: F = 1.220; *p* = 0.279), health profession (anxiety t = −1.694; *p* = 0.091; and t depression = −1.651; *p* = 0.099), job dismissal (t anxiety = −1.554; *p* = 0.121 and t depression = −1.546; *p* = 0.123) and having been diagnosed with SARS-CoV-2 infection (t anxiety = 0.104; *p* = 0.917 and t depression = 0.010; *p* = 0.992).

Having undergone diagnostic tests for COVID-19 had no statistical significance for anxiety (t = −1.226; *p* = 0.221), but it did for depression (t = −2.324; *p* = 0.020). However, having a family member or friend who had suffered from SARS-CoV-2 produced statistically significant differences in anxiety (t = −2.183; *p* = 0.029), but not in depression (t = −1.851; *p* = 0.065).

Finally, subjects who previously required psychological or psychiatric care were more likely to suffer from anxiety (t = −5.385; *p* = 0.000) and depression (t = −7.136; *p* = 0.000). Similar results were obtained if such help was needed at the time of the study (t anxiety = −9.144; *p* = 0.000 and t depression = −10.995; *p* = 0.000).

### 3.3. Regression Analysis

We did a successive step multiple linear regression analysis to determine which variables played a more important role in producing anxiety and depression symptoms (*p* < 0.01) (Table 5).
Total anxiety score = 10.226 + (1400 × cohabitation) − (0.03 × age) + (1.394 × COVID symptoms)(1)

**Theorem** **1.**
*The variables ‘living with someone during confinement’, ‘age’ and ‘presenting symptoms of COVID infection’ increase the score on the anxiety subscale of the GADS instrument and therefore the subjective perception of suffering from anxiety. The results determine a predictor model of anxiety (R^2^ = 0.245).*


Total depression score = 11.221 (−0.046 × age) + (1.281 × cohabitation)(2)

**Theorem** **2.**
*The variables ‘age’ and ‘living with someone during confinement’ increase the score on the depression subscale of the GADS instrument and therefore the subjective perception of suffering from depression, with a predictive model of depression (R^2^ = 0.289).*


## 4. Discussion

Taking into account the research findings, the working hypothesis can be partially accepted, since the factors age, gender, marital status, suffering from symptoms compatible with COVID-19 and requiring previous or current psychological care, were related to having an increased risk of anxiety and/or depression. Having undergone a diagnostic test for SARS-CoV-2 increased the risk of depression whereas having a family member or friend sick with COVID-19 increased the risk of anxiety.

Based on the results, it seems logical to say that confinement has had a negative effect on the population of central Spanish areas, which has been at risk of suffering from symptoms of anxiety and depression, results comparable with other studies carried out in Spain [3,5] and other countries [13,14,15,16,17,18,19].

It is surprising that unemployment was not identified as an essential factor for either depression or anxiety, since it is usually related to symptoms thereof in other studies [5,7,15,18,24]. The reason for this result is unknown, however, it may be related to participants responding more positively to a temporary dismissal that resulted from a reversible adjustment of employment levels due to the pandemic. Similarly, ample scientific literature describes the risk of suffering from anxiety and depression in the health professions [1,7,25]. In both cases, the low representation of these groups in the sample may have influenced the results. The economic influence of the pandemic should be carefully monitored by public organizations to support financially and psychologically numerous self-employed professions [26].

The low rates of positive COVID-19 diagnoses are consistent with epidemiological data published by the Ministry of Health, according to which around 5% of the population had developed antibodies to SARS-CoV-2 at the time of study [27]. The frequency of symptoms compatible with SARS-CoV-2 was higher than the percentage of positive diagnostic tests, probably, as other authors point out, because these symptoms are sometimes nonspecific, as is the case with fever, dry cough, and gastrointestinal symptoms [28].

Unsurprisingly, having required psychological help previously or at the time of study was related to the presence of anxious–depressive symptoms. However, some participants had had professional help at some point in their life but not during the pandemic. The cause of this effect is unknown, and it should be studied in depth in future research, since it contradicts the existing literature [17,29,30].

That younger people are at greater risk of anxiety and depression than older people was not surprising, coinciding with Ahmeda et al. [14]. Age was the variable that most influenced the presence of anxious–depressive symptoms together with gender, as corroborated by other studies [13,15,18,19]. In China, the most affected age range is 21 to 40 years [14]. In India, 15 to 35 years of age [15]. In the United Kingdom, early ages are the most affected [18] and in Denmark the middle ages [19].

Having to undergo a diagnostic test for COVID-19 increases the risk of depression. This result agrees with that of Chinese researchers who related it to mechanisms of anticipation of suffering the disease [31]. In addition, having a close person diagnosed as COVID-19-positive was related to anxiety, as also pointed out by Inchausti et al. [7] and Alamri et al. [32].

Living with at least two people has been considered an advantage since it may provide emotional support [18], but not all the scientific literature agrees on this point. Some studies highlight the need for a space of solitude within your own home, which is more complicated if you live with someone [23,33]. Jimenez et al. agree on quality of cohabitation and age were found to be key variables in the psychological impact of confinement [34]. Our results show that, during confinement, the number of cohabitants was not relevant for the development of anxiety or depression. However, cohabitation in general was relevant, as it constituted a risk factor for suffering from anxiety and depression. Marital status was another variable that had an influence on presenting anxious–depressive symptoms, and is related to cohabitation. Some research has revealed a greater perceived social support by people who have a partner, which seems of utmost importance during confinement [18,23].

Finally, it should be noted that the study provides a reality that, in the early stages of the pandemic, when the absolute priority was the lives of people, was not taken into account. Although some authors have published studies on this subject [34], in the case of Spain there is not much research and even less in our population group belonging to the central region of the country. The findings of this research show the possibility of suffering from anxiety and depression in a situation of home confinement and justify the creation of health programs to prevent this alteration of psychological health, as well as early detection, follow-up and treatment of the problems generated by isolation in the home in the face of an illness that entails major changes in living and working habits. Certain skills such as self-efficacy and the possibility that increasing self-compassion may be used to promote better mental health in similar situations [35,36,37].

### Limitations

We encountered difficulties during the study due to the circumstances associated with its design, which is why we might have incurred a selection bias in the participants due to a non-randomized sample. We assumed this bias due to the impossibility of accessing a large sample volume in such short period of time with other sampling methods. Due to the time required for approval by the ethics committee and the different rates of de-escalation in the different autonomous communities, most of the sample came from Castilla y León and Madrid, which impedes generalizing the results to the entire country. This sampling, however, strengthens the homogeneity of our data since both regions were still in Phase 0 of the de-escalation on the dates of study. Based on these issues, we propose to obtain a representative sample of the national population in the future. In addition, the use of a Likert scale questionnaire may generate response biases such as social desirability and acquiescence response [38,39], validation data can help quantify or mitigate this issue [40,41].

## 5. Conclusions

This study aimed to assess the risk of anxiety and/or depression during confinement due to the COVID-19 pandemic in the Spanish population. We found that 63% of the sample was at risk of suffering from anxiety and 64.9% at risk of depression.

The factors age, gender, marital status, suffering from symptoms compatible with COVID-19 and needing previous or current psychological help, were related to a greater risk of anxiety and/or depression. The risk of anxiety was fundamentally related to cohabitation, age and presenting symptoms compatible with COVID-19, while the risk of depression was mainly related to cohabitation and age.

In conclusion, taking care of the mental health of the population is essential in situations of confinement due to a pandemic. This requires developing action plans that allow an immediate response in the event of a further wave of SARS-CoV-2 or pandemics caused by other infectious agents.

## Figures and Tables

**Table 1 ijerph-18-05732-t001:** Frequency distribution and percentages of general variables (* *N* = 808), Spain, 2020.

Variables	Values: Frequency (*N*)/Percentage (%)
Gender	
Male	158 (19.6)
Female	650 (80.4)
Age (years)	
<40	325 (40.2)
40–49	221 (27.4)
50–59	151 (18.7)
>60	111 (13.7)
Marital status	
Married	354 (43.8)
Partnered	73 (9.0)
Separated or divorced	63 (7.8)
Single	306 (37.9)
Widowed	12 (1.5)
Healthcare worker	
Yes	172 (21.3)
Autonomous community	
Castilla y León	619 (76.6)
Madrid	85 (10.5)
Other	104 (12.9)
Number of cohabitants	
2	270 (33.4)
3	202 (25.0)
4	203 (25.1)
More than 4	48 (5.9)
Lived alone	85 (10.5)
Dismissal from work	
Yes	60 (7.4)
COVID-19 diagnostic	
Yes	30 (3.7)
Diagnostic COVID-19 test	
Yes	109 (13.5)
COVID-19 symptoms	
Yes	151 (18.7)
Family member or friend with COVID-19	
Yes	395 (48.9)
Previous psychological help	
Yes	244 (30.2)
Current psychological help	
Yes	87 (10.8)
Confinement in the household (^†^ *n* = 727)	
Yes	254 (34.9)

* *N*—total number of participants in the sample; ^†^ *n*—number of participants that were evaluated.

**Table 2 ijerph-18-05732-t002:** Affirmative responses in anxiety and depression subscales of the GADS * (^†^ *N* = 808), Spain, 2020.

**Anxiety Subscale of the GADS ***	
**Key Symptom**	**Values:** **Frequency (*N*)/Percentage (%)**
Anguished and nervous	402 (49.8)
Worried	586 (72.5)
Irritable	377 (46.7)
Difficulty relaxing	435 (53.6)
Bad sleep quality	472 (58.4)
Headache	417 (41.6)
Tremor and/or tingling and/or dizziness and/or sweating and/or diarrhea	201 (24.9)
Worried about health	449 (55.6)
Difficulty falling asleep	472 (58.4)
**Depression Subscale of the GADS ***	
**Key Symptom**	**Values:** **Frequency (*N*)/Percentage (%)**
Low energy levels	464 (57.4)
Loss of interest in things	314 (38.9)
Loss of self-confidence	163 (20.2)
Hopelessness	188 (23.3)
Difficulties concentrating	422 (42.2)
Weight loss	150 (18.6)
Wakes up earlier than usual	369 (45.7)
Slowness in carrying out activities	383 (47.4)
Feeling worse in the morning	244 (30.2)

* Goldberg Anxiety and Depression Scale; ^†^
*N*—total number of participants in the sample.

**Table 3 ijerph-18-05732-t003:** Prevalence of anxiety and depression symptoms according to GADS * (*N* = 808), Spain, 2020.

Variables	Frequency (^†^ *n* = 808)	Percentage %	CI 95%
*Symptoms of anxiety*			
Yes (=4)	509	63	59.6–66.4
No (<4)	299	37	33.6–40.4
*Symptoms of depression*			
Yes (≥2)	524	64.9	61.5–68.1
No (<2)	284	35.1	31.8–38.4

* Goldberg Anxiety and Depression Scale; ^†^ *n*—number of participants that were evaluated.

**Table 4 ijerph-18-05732-t004:** Inferential analysis between variables and scale GADS ^†^ (*N* = 808), Spain, 2020.

**Variables**	**Pearson Correlation Coefficient (r)**	
	**Anxiety**	**Depression**
Age	−0.139 *	−0.153 *
Number of cohabitants	0.044	0.027
**Variables**	**ANOVA (F)**	
	**Anxiety**	**Depression**
Marital status	2.893 **	3.011 **
Cohabitation	13.636 **	10.007 **
Geographical place	0.934	1.220
**Variables**	**Student *t*-Test (*t*)**	
	**Anxiety**	**Depression**
Gender	−4.152 **	−4.178 **
COVID symptoms	−4.177 **	−3.791 **
Health profession	−1.694	1.651
Job dismissal	−1.554	−1.546
COVID diagnostic	0.104	0.010
COVID tests	−1.226	−2.324 *
COVID diagnostic of family member or friend	−2.183 *	−1.851
Psychological or psychiatric care (pre pandemic)	−5.385 *	−7.136 **
Psychological or psychiatric care (during pandemic)	−9.144 **	−10.995 **

* *p* < 0.05; ** *p* < 0.01; ^†^ Goldberg Anxiety and Depression Scale.

**Table 5 ijerph-18-05732-t005:** Models of regression analysis. Theorem 1 and Theorem 2 (*N* = 808), Spain, 2020.

**Factor**	**Theorem 1**					
	**B**	**Standard Error**	**β**	***t***	***p***	**CI 95%**
Constant	10.226	1.077		9497	0.000	8.102–12.35
Cohabitation	1.400	0.301	0.322	4.658	0.000	0.807–1.003
Age	−0.033	0.010	−0.225	−3.259	0.001	−0.052–−0.013
COVID symptoms	1.394	0.447	0.197	3.118	0.002	0.512–2.275
**Factor**	**Theorem 2**					
	**B**	**Standard Error**	**β**	***t***	***p***	**CI 95%**
Constant	11.221	0.867		12.944	0.000	9.511–12.931
Age	−0.046	0.009	−0.333	−4.968	0.000	−0.064–−0.028
Cohabitation	1.281	0.279	0.308	4.600	0.000	0.732–1.830

## Data Availability

The data presented in this study are available on request from the corresponding author.

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
