# Peer review of "Evaluation of the Risk of Anxiety and/or Depression during Confinement Due to COVID-19 in Central Spain"

_ijerph, 2021, doi:10.3390/ijerph18115732_

Round 1

Reviewer 1 Report

This is an important and interesting study concerning SARS-Cov19 problems in central Spain. Article is very-well written and a study sample is great. Yet, anxiety and depression during confinement problems are on the rise. Studies of this kind are on demand, however some minor criticism should be raised:

Materials and methods

L70 - Please clarify inclusion and exclusion criteria, if any

Discussion section

This section is a little short, as there were a lot more articles concerning first wave of the pandemic released recently which authors did not mention. Therefore much more articles from European countries authors may discuss with, for instance https://www.mdpi.com/1660-4601/18/3/1281 I feel that incorporating few more references and expanding its' list to approx. 40 will surely improve an overall manuscript quality and eventually attract more readers. 

Good luck!

Author Response

Reviewer 1

We wish to thank you all for your constructive comments in this round of review. Your comments provided valuable insights to refine its contents and analysis. In this document, we try to address the issues raised as best as possible.

As you suggest, we have included in lines 80-86 the inclusion and exclusion criteria of the sample. We believe that your suggestion makes the sampling clearer.

We accept the reviewer's suggestion; we agree that expanding the bibliography of the article can increase its visibility. Therefore, we have increased the number of citations as you indicated and we have included the reference suggested by you (reference number 25).

With sincere appreciation,

The authors.

Reviewer 2 Report

The authors aimed to study the most relevant proximal/distal factors associated with the emotional wellness of middle-aged (43 ± 19y) Spanish adults during the COVID-19 lockdown (first wave, 4-20 May 2020). After conduction, a cross-sectional/observational study with 808 participants [ mostly married (44%) women (80%) from Castilla y Leon (77%)] non-testing positive to SARS-CoV-2 (87%) nor having symptoms (81%)]. Most participants (47-73%) referred to be worried (general and about health), having difficulties sleeping and relaxing, low energy, and slowness to perform daily activities, and 6 out of ten exhibited anxiety and/or depression. (A) Gender, (B) cohabitation & (C) having symptoms of COVID-19 (C) (p<0.0001), (D) marital status (p= 0.002) and (E) age (p= 0.01), were strongly associated with anxiety [stepwise linear regression (SLR): B+C+E] and depression (SLR: E+B). This reviewer considers that the experimental design, survey execution, and results are valuable, although mild changes could still improve its comprehension and scientific soundness:

  1. The manuscript must be re-reviewed by a formal translation agency or native English-speaking colleague.
  2. Introduction: A) I believe that some prevalence data and/or socio-behavioral features observed in studies 13-19 should be included briefly. B) It should also be stated (shortly) why it was decided to use Bayesian networks to study the complex relationship between observed variables, as well as some characteristics of the experimental design that are relevant.
  3. Materials & Methods: A) Sections 2.2 and 2.6 should be put together, B) Please include the self-administered questionnaire as supplementary material, C) Please support with one or two references the scientific validity of self-report studies of disease/risk factors in middle-aged populations, since they are commonly biased (Doi: 10.1007/978-94-007-0753-5_4046, 3389/fpsyg.2019.02309) as compared to the same type of surveys in older adults.
  4. Results & Tables. A) Table 1: Reduce its length by removing all negative ("no") values (paired data) and merge frequency and percentage columns into one column by describing data as n (%); see the following as example (Table 1: https://www.ncbi.nlm.nih.gov/pmc/articles/PMC7588260/pdf/formative_v4i10e22043.pdf ). Include age as stratified frequencies (<40, 40-49, 50-59, >60y), B) Table 2: Rearrange horizontally [Columns: (i) Anxiety symptom, (ii) n (%), (iii) Depression symptom, (iv) n (%)], eliminate “item” column. C) Include a new Table 3 depicting both significant and non-significant determinants of depression (having vs. not having) and anxiety (having vs. not having) including F, t or χ2 values or just p-values as is customary in epidemiological studies (e.g., https://files.eric.ed.gov/fulltext/EJ1196202.pdf , https://www.revistapcna.com/sites/default/files/012.pdf ).C) Suggestion: Data depicted and discussed in section 3.3. could be transformed into a figure (Venn diagrams?, Bayesian maps?), since anxiety and depression show practically the same social determinants (prodromal relationship between them?).
  5. Discussion. The authors could take advantage of the fact that nowadays there are many studies carried out worldwide relating COVID-19 lockdown with the emotional state of adults, children, and the elderly. Please include at least two studies that support your results.

Author Response

We wish to thank you all for your constructive comments in this round of review. Your comments provided valuable insights to refine its contents and analysis. In this document, we try to address the issues raised as best as possible.

We would like to point out that the manuscript has been translated by a bilingual person as attested by a report of the services contracted for this purpose.

In accordance with your suggestion, data from references 13 to 19 are included in the introduction (lines 59-68).

The statistical analysis carried out in the study has been further specified in the material and method section. (lines 130-131).

Sections 2.2 and 2.6 (lines 87-95) have been joined.

The questionnaire used to collect dependent variables is included as supplementary material. In addition, participants answered 18 anxiety and depression-related questions from The Goldberg Anxiety and Depression Scale (GADS), which are copyrighted and cannot be disseminated without permission from the authors who created and validated the questionnaire for this purpose.

Your suggestion has been taken into account, although we consider it more appropriate to add such information within the limitations of the study (lines 294-297), including citations to the references you mention (reference number 37 and 39).

Table 1 is reduced by eliminating negative responses and restructured by placing frequencies and percentages in the same column. In addition, age is included in the format that specifies ranges (line 152).

The items in table 2 are removed and restructured as in the case of table 1, unifying frequencies and percentages in a single column (line 164-165).

Table 4 is created to specify the significance values ​​r, t, and F, thereby clarifying the inferential results (line 203).

Section 3.3. is clarified with a new table, table 5, according to the suggestion of another reviewer. We hope the table is correct for you (line 216).

We have included in the discussion more recent studies, some of which deal with the age of the population. In addition, studies supporting our results are also included (lines 236-238, 253-255, 259-260, 264-266, 273-283).

With sincere appreciation,

The authors.

Reviewer 3 Report

First of all, I wanted to congratulate the authors for the article, since I find it very interesting to provide data on mental health in a country with a different model of confinement from other countries. However, the article has some shortcomings that need to be addressed:

  • The numerical explanation of the results in the abstract is not very clear.

  • The Introduction is too short, since there are a lot of studies on the psychological impact of quarantine, even in Spain.

  • It would be advisable to include this study (“Psychological Impact of COVID-19 Confinement and Its Relationship with Meditation”) in the Introduction and/or the Discussion, since it speaks about the psychological impact of quarantine (depression, anxiety, stress) in Spain and includes variables such as the quality of cohabitation. This study was published in this same journal: https://www.mdpi.com/1660-4601/17/18/6642

  • It is not very clear the rationale of the article. What knowledge this article adds to the previous scientific literature on this subject?

  • Why did you use only one questionnaire in the evaluation of the psychological impact of quarantine?

  • Please, include the table with the complete data of the regression analysis.

Author Response

We wish to thank you all for your constructive comments in this round of review. Your comments provided valuable insights to refine its contents and analysis. In this document, we try to address the issues raised as best as possible.

Thank you very much for your comment, the error detected in the results part of the abstract (line 25) is corrected.

The length of the introduction has been increased (lines 59-68). In addition, we have included the suggested citation in the discussion (reference number 33), also increasing the number of articles cited to improve the quality of our work.

With regard to what the article contributes to science, we can say that our study contributes a reality that, in the early stages of the pandemic, when the absolute priority was people's lives, was not taken into account. Although some authors have published studies on this subject, in the case of Spain there is not much research and even less in our population group living to the central region of the country. The findings of this research highlight the possibility of suffering from anxiety and depression in a situation of home confinement and justify the creation of health programs to prevent this alteration of mental health, as well as early detection, monitoring and treatment of the problems generated due to isolation at home in the face of a disease that entails major changes in living and working habits. This has been included in the discussion (lines 273-283).

Regarding the use of a single study questionnaire, it was due to the fact that at the time of data collection time was pressing to collect answers to the study question, opting for the Goldberg Anxiety and Depression Scale (GADS), as it is a highly reliable instrument that is perfectly suited to our study objectives. In addition, we found that this questionnaire has also been used regularly by other authors in other catastrophic situations. It must be taken into account that, during the data collection period, the Spanish population was asked by many working groups to respond to questionnaires of various kinds, so we thought that using a short questionnaire would induce the population to respond to it more than if several measuring instruments were used.

Table 5 has been included with the data from the regression analysis, according to your suggestion (line 216).

With sincere appreciation,

The authors.

Round 2

Reviewer 3 Report

The way to express the results in the summary remains unclear. For example, although the age or gender variable is said to have been statistically significant, it does not specify in which age group or gender shows more anxiety or depression.

In the introduction to the article, it should be specified why it is important to study the central area of ​​Spain with respect to other studies where confinement has been studied in Spain in general.

In the Annex the answers for "Yes" and "No" do not all have the same indentation in each answer.

Author Response

At this point we regret not being able to improve the results, as the statistical tests used on age and gender do not allow us to make these distinctions, but we are already working on a second investigation that will allow us to specify these aspects with robust statistical tests.

About this aspect, we fully agree with you, and have therefore incorporated a paragraph in the introduction specifying this point (lines 68-71).

The reference for this aspect is: Pollán, M.; Pérez-Gómez, B.; Pastor-Barriuso, R.; Oteo, J.; Hernán, M.A.; Pérez-Olmeda, M.; Sanmartín, J.L.; Fernández-García, A.; Cruz, I.; Fernández de Larrea, N.; et al. Prevalence of SARS-CoV-2 in Spain (ENE-COVID): a nationwide, population-based seroepidemiological study. Lancet. 2020, 396(10250), 535–544. doi:10.1016/S0140-6736(20)31483-5

With regard to the error in the indentation of the answers in the annex, we very much regret the error and correct it as you suggest.
